# Aspirin and Primary Cancer Risk Reduction in Ischemic Cardiac or Cerebrovascular Disease Survivors: A Nationwide Population-Based Propensity-Matched Cohort Study

**DOI:** 10.3390/cancers15010097

**Published:** 2022-12-23

**Authors:** Yen-Hsiang Liao, Ren-Jun Hsu, Tzu-Hwei Wang, Chen-Ta Wu, Sheng-Yao Huang, Chung-Y. Hsu, Wen-Lin Hsu, Dai-Wei Liu

**Affiliations:** 1Department of Radiation Oncology, Hualien Tzu Chi Hospital, Buddhist Tzu Chi Medical Foundation, Hualien 970473, Taiwan; 2School of Medicine, Tzu Chi University, Hualien 970374, Taiwan; 3Cancer Center, Hualien Tzu Chi Hospital, Buddhist Tzu Chi Medical Foundation, Hualien 970473, Taiwan; 4Department of Radiation Oncology, Taichung Tzu Chi Hospital, Buddhist Tzu Chi Medical Foundation, Taichung 427213, Taiwan; 5Graduate Institute of Biomedical Sciences, College of Medicine, China Medical University, Taichung 406040, Taiwan

**Keywords:** aspirin, secondary prevention, ischemic cardiac disease, ischemic cerebrovascular disease, primary cancer risk

## Abstract

**Simple Summary:**

Long-term low-dose aspirin use was associated with a reduced risk of primary cancer in survivors of ischemic cardiac or cerebrovascular disease. Thus, in contrast to the situation for the general population (for which the anticancer effects of aspirin are still controversial), long-term low-dose aspirin use in these patients, though originally employed for the secondary prevention of ischemic attack, also has extra anticancer benefits.

**Abstract:**

Ischemic cardiac or cerebrovascular disease (ICCD) survivors represent a subpopulation with a high cancer risk. Antiplatelet medications, such as aspirin, remain a fundamental therapy for the secondary prevention of ischemic attack in these patients. We conducted a population-based cohort study to investigate the association of long-term low-dose aspirin use with the risk of primary cancer in ICCD survivors. Patients aged ≥20 years with newly diagnosed ICCD (*n* = 98,519) between January 2000 and December 2013 were identified from the Taiwan National Health Insurance Research Database. The aspirin user and nonuser groups (each *n* = 24,030) were propensity-matched (1:1) for age, sex, comorbidities, prior medications, ICCD diagnosis year, and year of index dates. The incidence rate of primary cancer was significantly lower in the user group (6.49/1000 person-years) than in the nonuser group (14.04/1000 person-years). Multivariate Cox regression analysis indicated that aspirin use was an independent factor associated with a reduced risk of primary cancer (aHR (95% confidence interval) = 0.42 (0.38–0.45)) after adjustment. Kaplan–Meier curve analysis revealed that the cumulative incidence rate of primary cancer was significantly lower (*p* < 0.0001) in the user group than in the nonuser group over the 14-year follow-up period. Subgroup analyses demonstrated that this anticancer effect increased with duration of treatment and with similar estimates in women and men. In addition, aspirin use was associated with a reduced risk for seven out of the ten most common cancers in Taiwan. These findings suggest the anticancer effect of aspirin in ICCD survivors and provide information for assessing the benefit-to-risk profile of aspirin as an antiplatelet medication in these patients.

## 1. Introduction

Aspirin is a nonsteroidal anti-inflammatory drug that has potent antiplatelet properties. Prophylactic low-dose aspirin use has been recommended for the primary prevention of cardiovascular diseases (CVD) in adults aged 40–59 years [1] or up to 70 years [2] who are at a high risk for CVD. During the past few decades, regular long-term aspirin intake has been studied as a potential chemopreventive approach for primary cancer prevention in the general population [3,4]. Antiplatelet actions and several additional mechanisms have been proposed to contribute to the anticancer effect of aspirin [5,6]. However, whether low-dose aspirin has anti-cancer benefit is still controversial [3,4,7,8,9,10,11,12,13]. Specifically, low-dose aspirin use has been found to be associated with a reduced risk of total cancer [7], gastric cancer [14,15], esophageal cancer [14,15], pancreatic cancer [14,16], ovarian cancer [14,17], breast cancer [14,18], prostate cancers [14,19], head and neck cancer [20], liver cancer [21], lung cancer [22], and most notably colorectal cancer (CRC) [4,23,24]. Conversely, low-dose aspirin use has also found to have null effect on the risk of total cancer [8,10,12], bladder cancer [9,12], breast cancer [9,12], esophageal cancer [9,12], gastric cancer [9,12], pancreatic cancer [9,12], uterine cancer [9,12], and CRC [11,12]. While the anticancer effect of aspirin in the general population remains a topic of debate, there is growing interest to investigate this effect in subpopulations with a high cancer risk, such as patients with familial adenomatous polyposis [25], hepatitis B or C infection [21,26], or diabetes [27,28].

Survivors of ischemic cardiac or cerebrovascular disease (ICCD) represent another subpopulation with a high cancer risk, because CVD and cancer have an extensive overlap in risk factors and some common basic molecular pathways for pathogenesis [29]. Indeed, patients with various types of cancer have a high prevalence of preexisting CVD [30]. Antiplatelet medications, including aspirin, remain a fundamental therapy for the secondary prevention of ischemic attack in ICCD survivors [31,32]. In patients with ICCD treated with long-term low-dose aspirin, the reduced level of risk for recurrent heart attack or stroke generally outweighs the increased risk of bleeding [33,34,35]. The United Kingdom transient ischemic attack (UK-TIA) aspirin trial [36] and Swedish Aspirin Low-Dose Trial (SALT) [37] were two studies aiming for secondary prevention of cerebrovascular ischemic events. Using follow-up data from patients in UK-TIA, daily use of aspirin was found to be associated with a reduced risk of CRC [23] and cancer-related death [38,39]. Conversely, using follow-up data from patients in UK-TIA and SALT, daily use of aspirin was found to have null effect on CRC incidence risk [40] and CRC-related mortality [41,42]. As such, the pleiotropic effect of aspirin against cancer in ICCD survivors remains largely unclear. This topic is clinically important because the findings may provide information for assessing the benefit-to-risk profile of aspirin as an antiplatelet medication in patients with ICCD.

The objective of the current study was to investigate the association of low-dose aspirin use with the risk of primary cancer in ICCD survivors. We used a nationwide population-based dataset to conduct a propensity score-matched cohort study to achieve this goal. Patients were classified as aspirin users and aspirin nonusers for subsequent univariate and multivariate analysis.

## 2. Materials and Methods

### 2.1. Data Source

A population-based propensity-matched cohort study was conducted using the Taiwan National Health Insurance Research Database (NHIRD). The National Health Insurance program provides the entire population of Taiwan with universal health insurance and covers 99% of their health care needs and medical services. NHIRD includes ambulatory care, inpatient care, and registration data of the insured [41]. Ambulatory care claims contain the individual dates of visits, as well as their International Classification of Disease, Ninth Revision, Clinical Modification (ICD-9-CM) codes. Their validity and accuracy have been demonstrated in previous studies [41]. In this study, we used the longitudinal health insurance database 2000 (LHID 2000). LHID 2000 contains 1,000,000 participants (approximately 5% of Taiwan’s population) randomly selected from National Health Insurance beneficiaries in Taiwan [41]. We collected data regarding basic demographic factors, comorbidities, and history of medication. The data sampling from the LHID 2000 dataset is representative of the whole population during this time frame.

### 2.2. Study Cohort and Design

We searched the records in NHIRD from January 2000 to December 2013 to identify patients aged ≥20 years with newly diagnosed ICCD. Patients with ICCD were identified by ICD-9-CM codes (410–414 for ischemic cardiac disease and 433–436 for ischemic cerebrovascular disease). To enhance diagnostic validity, patients with at least two consistent diagnoses of ICCD from outpatient medical records were enrolled (Figure 1). Among the patients with a new diagnosis of ICCD, those who were given a prescription for aspirin (defined as >90 days of daily doses), either immediately or after a period of time, were defined as aspirin users, whereas those without such as prescription were defined as aspirin nonusers. In Taiwan, low-dose aspirin is a prescription drug with 100 mg/day as the most frequent formulation indicated for CVD prevention [24]. We set the first date on which aspirin was prescribed as the index date for aspirin users, and matched the same interval for aspirin nonusers as their index date. To reduce the bias due to confounding factors, the aspirin users and nonusers were propensity-matched for age, sex, comorbidities, medications, diagnosis year of ICCD, and year of index dates to achieve a 1:1 balanced cohort. These two study cohorts were matched using age intervals of 20–39, 40–59, 60–79, and ≥80 years. This study was approved by the Institutional Review Board of China Medical University Hospital CMUH104-REC2-115 (CR-2). For this retrospective study, informed consent was waived in accordance with the institutional guidelines.

### 2.3. Covariate Assessment

Basic demographic data included age and sex. Comorbidities, including hypertension, diabetes mellitus, hyperlipidemia, atrial fibrillation, alcohol-related illness, mild liver disease, moderate or severe liver disease, hepatitis B or C, rheumatic disease, myocardial infarction, congestive heart failure, chronic pulmonary disease, peptic ulcer disease, gastrointestinal bleeding, and renal disease, were identified by corresponding ICD-9-CM codes. To reduce coding errors, a comorbidity was defined as the presence of situations when ICD-9-CM codes appeared at least two times in the outpatient records, or once in inpatient claims, before the cohort entry date of any patient enrolled in this study. Medications that are commonly prescribed to this patient population within 6 months prior to the index date were also identified. Drug prescriptions, based on Anatomical Therapeutic Chemical (ATC) codes, were also registered in the database. These medications included antihypertensive agents, drugs for cardiac therapy, peripheral vasodilators and vasoprotectives, 3-hydroxy-3-methylglutaryl coenzyme A reductase inhibitors, other lipid-modifying agents, hypoglycemia agents, coumadin and heparin, other antithrombotic agents, proton pump inhibitors, histamine type-2 receptor antagonists, antacids, estrogens and progesterone, and nonsteroidal anti-inflammatory drugs. No history of aspirin use in the study participants was found during this period. However, we cannot exclude the possibility that these participants used aspirin before this period and cannot define the aspirin users in this study as new users.

### 2.4. Study Outcome

The study outcome was the event of newly diagnosed cancer as identified by ICD-9-CM codes following the index dates. Data on the 10 most common cancers in Taiwan were collected, and these cancers included head and neck cancer, esophageal cancer, stomach cancer, colon cancer, hepatoma, pancreatic cancer, lung cancer, breast cancer, ovary cancer, and prostate cancer [42]. The follow-up for patients began on the index date and lasted until withdrawal from the National Health Insurance program, until the diagnosis of primary cancer, or on 31 December 2013, whichever occurred first.

### 2.5. Statistical Analyses

Categorical variables were presented as *n* (%) and were compared using the Chi-squared test. The balance of the covariates was measured by using standardized mean difference. Univariate and multivariate Cox proportional hazard regression models were used to calculate crude hazard ratios (HRs) and adjusted HRs (aHRs), respectively, with 95% confidence intervals (CIs) after adjusting for potential confounding variables. HRs adjusted for aspirin use, age, sex, baseline comorbidities, and drug use in the Cox proportional hazard model. To estimate the cumulative risk of primary cancer in the two study cohorts, we employed the Kaplan–Meier method with significance based on the log-rank test. All statistical analyses were performed using SAS version 9.4 (SAS Institute Inc., Cary, NC, USA) and R software (R Foundation for Statistical Computing, Vienna, Austria). A *p* value of <0.05 was considered statistically significant.

## 3. Results

### 3.1. Demographic and Baseline Clinical Characteristics

We identified 98,519 patients with new diagnoses of ICCD during the study period. After exclusion, 53,047 aspirin users and 38,353 nonusers were identified. After propensity score matching, each study group consisted of 24,030 patients (Figure 1). The mean age of the study cohort was 60.27 years, and the majority were men. Low-dose aspirin is a prescription drug in Taiwan with a dosage of 100 mg/day for ICCD survivors, and the mean duration of aspirin use in our cohort was 184 ± 383 days (mean ± SD). Table 1 shows the demographic and baseline clinical data of the two matched cohorts. As shown, all standardized mean differences in the covariates were less than 0.2, indicating a negligible difference between these two study cohorts (i.e., the two cohorts are well-balanced).

### 3.2. Risk of Developing Primary Cancer in Aspirin Users versus Aspirin Nonusers

The incidence rate of primary cancer in the aspirin user group was 6.49 per 1000 person-years, which was lower than that in the aspirin nonuser group (14.04 per 1000 person-years). The average follow-up durations for the aspirin user and nonuser groups were 5.49 and 3.15 years, respectively. Univariate Cox regression analysis revealed that aspirin users had a significant decrease in HR for primary cancer (crude HR (95% CI) = 0.48 (0.44–0.52)) compared with aspirin nonusers. Multivariate Cox regression analysis indicated that aspirin use remained an independent factor associated with a reduced risk of primary cancer (aHR (95% CI) = 0.42 (0.38–0.45)) after adjustment for confounding factors. The Kaplan–Meier curve analysis revealed that the cumulative incidence rate of primary cancer in the aspirin user group was significantly lower (Figure 2, log-rank test, *p* < 0.0001) than that in the aspirin nonuser group over the follow-up period. Further analyses revealed the reduced risk of primary cancer in aspirin users with the use durations of <1 year (aHR (95% CI) = 0.43 (0.39–0.47)), 1–2 years (aHR (95% CI) = 0.37 (0.30–0.47)), 2–3 years (aHR (95% CI) = 0.42 (0.31–0.56)), and >3 years (aHR (95% CI) = 0.31 (0.23–0.43)) compared with aspirin nonusers (Table 2). Due to the strategy of identifying aspirin users, we were not able to confirm whether the aspirin users in this study were new users or if those in the final cohort were current users.

### 3.3. Subgroup Analyses for the Risk of Developing Primary Cancer in Aspirin Users versus Aspirin Nonusers

Subgroup analyses stratified by sex and age (Table 3) revealed that aspirin use was an independent factor associated with a reduced risk of primary cancer in females (aHR (95% CI) = 0.43 (0.37–0.50)) and males (aHR (95% CI) = 0.42 (0.38–0.47)) and in patients aged 40–59 (aHR (95% CI) = 0.45 (0.38–0.53)), 60–79 (aHR (95% CI) = 0.39 (0.35–0.44)), and ≥80 years (aHR (95% CI) = 0.43 (0.31–0.58)), but not in patients aged 20–39 years (aHR (95% CI) = 0.78 (0.39–1.54)). Additional subgroup analyses stratified by the 10 most common cancers in Taiwan (Table 4) revealed that aspirin use was an independent factor associated with a reduced risk of primary cancer of the head and neck (aHR (95% CI) = 0.42 (0.31–0.57)), esophagus (aHR (95% CI) = 0.45 (0.23–0.88)), stomach (aHR (95% CI) = 0.43 (0.30–0.62)), colon (aHR (95% CI) = 0.36 (0.31–0.43)), hepatoma (aHR (95% CI) = 0.51 (0.41–0.63)), lung (aHR (95% CI) = 0.55 (0.44–0.69)), breast (aHR (95% CI) = 0.42 (0.30–0.58)), and prostate (aHR (95% CI) = 0.33 (0.26–0.41)), but not for primary cancer of the pancreas (aHR (95% CI) = 0.84 (0.50–1.40)) or ovary (aHR (95% CI) = 0.75 (0.29–1.93)). The forest plot (Figure 3) schematically showed the reduced risk of developing these types of primary cancer associated with aspirin use. We have performed additional analysis of patients who had aspirin use within 90 days after ICCD. The data are shown in Appendix A. As shown, the analysis also showed lower cancer risk in aspirin users.

## 4. Discussion

The major finding of this population-based propensity-matched cohort study is that low-dose aspirin use was independently associated with a decreased risk of primary cancer in ICCD survivors after adjustment. Whether regular long-term aspirin intake is a chemopreventive approach for primary cancer prevention in the general population has been a much-debated topic [3,4,7,8,9,10,11,12,13,14,15,16,17,18,19,20,21,22,23,24]. ICCD survivors represent a subpopulation with a high cancer risk because CVD and cancer possess several similar risk factors and mechanisms of pathogenesis [29]. Patients diagnosed with various common cancers have a higher prevalence of preexisting CVD than the general population [30]. Given that preexisting CVD is associated with the subsequent development of cancer, it would be of clinically important to know that any drug for the treatment of CVD also has potential for primary cancer prevention. To this end, taking low-dose aspirin daily in ICCD survivors is not for the sake of primary cancer prevention, but for the secondary prevention of ischemic attack [31,32]. Although several antiplatelet therapies are available for these patients, according to current guidelines aspirin remains the antiplatelet agent of choice for the purpose of secondary prevention [43,44]. In ICCD survivors treated with long-term low-dose aspirin, the reduced level of risk for recurrent heart attack or stroke generally outweighs the increased risk of bleeding [33,34,35]. Accordingly, our finding regarding the extra anticancer effect may provide information for assessing the benefit-to-risk profile of aspirin as an antiplatelet medication in patients with ICCD.

Research on the anticancer effect of aspirin use in ICCD survivors is limited. Using data from UK-TIA trial patients with a scheduled treatment duration of 5 years or more, Flossmann et al. [23] reported that aspirin resulted in a significant reduction in the incidence of CRC during long-term follow-up. Using data of the same trial patients, Rothwell et al. [38] and Chan et al. [39] reported that aspirin decreased the odds of death due to cancer, but Guirguis-Blake et al. [40] demonstrated null effect of aspirin use on CRC incidence risk and CRC-related mortality. Using data from UK-TIA and SALT trial patients, Rothwell et al. [45] showed no effect of aspirin on long-term risk of death due to CRC. No study has been conducted to examine the anticancer effect of aspirin in patients with established ischemic cardiac disease. A retrospective cohort study [46] indicated that compared with nonusers, patients using aspirin as antiplatelet therapy for unspecified medical conditions had a lower risk of cancer during long-term follow-up. Thus, this large-scale cohort study provides clinical evidence that ICCD survivors initiating aspirin treatment may have a significantly lower risk for primary cancer development than their nonuser counterparts.

The findings from our subgroup analyses are generally consistent with those reported previously in the general population. We found that the anticancer effect of low-dose aspirin appears to be prominent when the aspirin use duration was 3 or more years, 2–3 years, 1–2 years, or less than 1 year, compared with aspirin non-use. Moreover, this anticancer effect was apparent for both men and women. These findings suggest that the benefit of aspirin against primary cancer increased with the duration of treatment [7,38,45] and with similar estimates in women and men [4,7]. Rothwell et al. [7] analyzed the data of patients from trials in primary CVD prevention and reported that allocation to aspirin reduced the overall cancer incidence from 3 years onward similarly in women and in men. Lin et al. [24] analyzed data from the Taiwan NHIRD and reported that low-dose aspirin use for durations of >1, 3, or 5 years was associated with decreased odds of developing CRC. We also found that this anticancer effect was apparent for patients aged above 40 years. Age is always an important factor in the study of the anticancer effects of aspirin because ICCD survivors of younger ages may have less vulnerability to developing cancer compared with those of older ages. A recent study [12] reported that daily low-dose aspirin had no effect on the incidence of overall cancer or various types of cancer in relatively healthy adults aged 70 years or older. If fact, the authors [12] concluded that, in older adults, aspirin treatment had an adverse effect on the later stages of cancer evolution. This conclusion was based on the findings that aspirin was associated with an increased risk of incident cancer that had metastasized or was stage 4 at diagnosis. Perhaps healthy older adults may have less vulnerability to the development of cancer compared with ICCD survivors. We additionally found that ICCD survivors who were exposed to low-dose aspirin were associated with a reduced risk of seven out of the ten most common cancers in Taiwan [42]. In the general population, aspirin use has been shown to reduce the risk of cancer in the head and neck [20], stomach [14,15], colon [23,24], hepatoma [21], lung [22], breast [14,18], and prostate [14,19]. The anticancer effects of aspirin on pancreatic cancer [14,16] and ovarian cancer [14,17] were not observed in this study. Several limitations must be considered in this study. First, our retrospective study was subject to several biases, including data collection and the differences between the two study cohorts. We believed that these biases could be minimized by the study design using propensity-matched analyses. However, unmeasured differences between these two cohorts may still account for our observed results and the confounding of immortal time bias should be considered. Covariates such as disease severity, smoking, lifestyle, and education may be related to cancer development. Unfortunately, the health insurance database used in this study does not register these data. Second, the definition of aspirin use was based on the prescription registered in the database. The exact aspirin regimen could not be collected from the database, but a low dose of 100 mg/day is the most frequent formulation indicated for CVD prevention in Taiwan [24]. In addition, aspirin is a prescription drug in Taiwan for this patient population and is not available over the counter; its prescriptions can be captured in the database. Third, findings from our subgroup analyses should be interpreted with caution as stratification results in limited statistical power. Fourth, our study design involved cohort-matching processes that inevitably exclude participants whose matching covariates (either one or several) did not qualify to be included into the analyzed cohort. Our findings should not be generalized to all patients. Fifth, the follow-up time for the study is very short (only 5.49 and 3.15 years for aspirin users and nonusers, respectively). With a short follow-up period, it is hard to say whether aspirin is associated with reduced risk of cancer, or whether aspirin simply delays cancer detection and diagnosis. Sixth, we initially identified *n* = 53,047 aspirin users and *n* = 38,353 nonusers, but our design used propensity score matching to generate two cohorts with the same sample size (*n* = 24,030) that are balanced 1:1 for age, sex, comorbidities, medications, diagnosis year of ICCD, and year of index dates. This matching process inevitably excludes *n* = 29,017 aspirin users and *n* = 14,323 nonusers whose matching covariates (either one or several) did not qualify to be included into the balanced cohort. Seventh, our results showed null associations for ovarian and pancreatic cancer, which may be due to the possibility of low power regarding subgroup analyses and a small sample size. Perhaps future investigations using in vitro or in vivo models are warranted to evaluate the effects of aspirin on pancreatic cancer, because its ability to act on multiple molecular targets in this cancer type has been proposed [47]. Eighth, our study enrolled East Asian people, an ethnically homogenous population. It is possible that there are racial differences in the metabolism of aspirin, which may explain some of the divergent results in this population compared to Western populations. This possibility remains to be explored in future studies.

## 5. Conclusions

In summary, low-dose aspirin use in ICCD survivors may reduce the risk of developing overall primary cancer and seven out of the ten most common cancers in Taiwan, or may delay the cancer detection and diagnosis. This anticancer effect of aspirin increases with duration of treatment and is apparent for patients aged 40 years or older, with similar estimates in women and men.

## Figures and Tables

**Figure 1 cancers-15-00097-f001:**
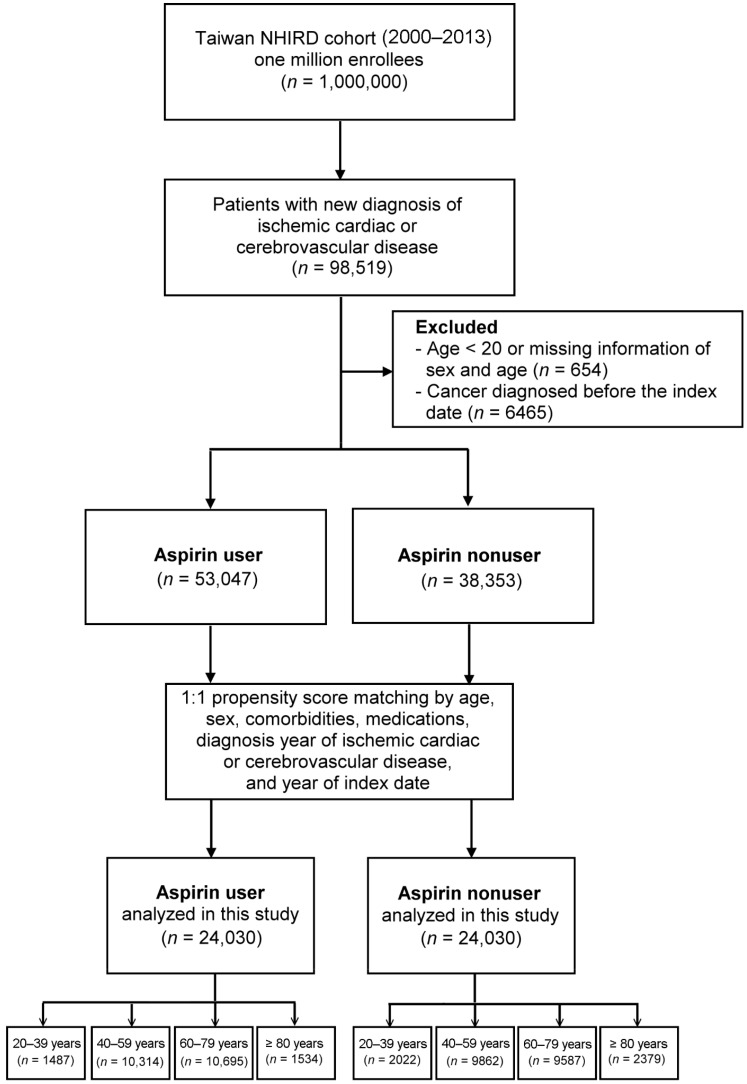
Flowchart of patient selection. NHIRD, National Health Insurance Research Database.

**Figure 2 cancers-15-00097-f002:**
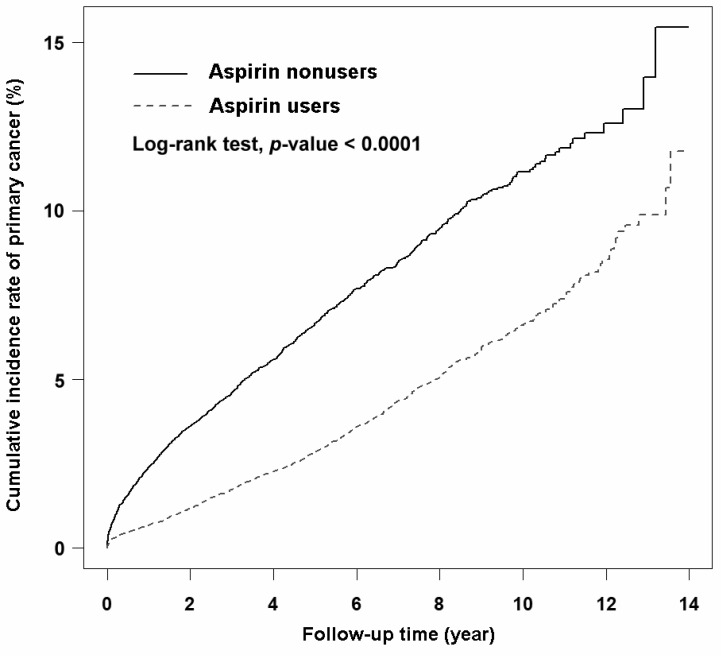
Kaplan–Meier curves showing the cumulative incidence of primary cancer in aspirin users and aspirin nonusers in this study.

**Figure 3 cancers-15-00097-f003:**
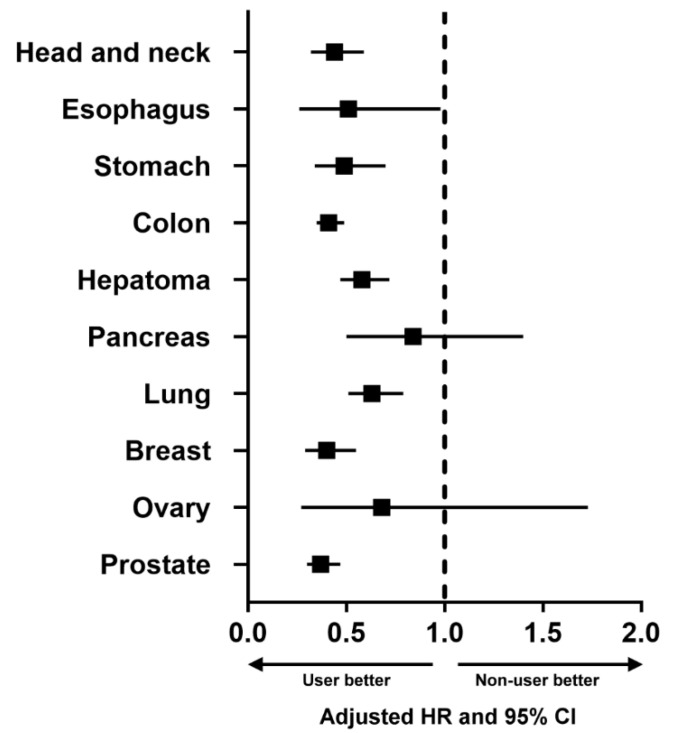
Forest plot showing the adjusted hazard ratio (HR) and 95% confidence interval (CI) of the 10 most common cancers in Taiwan, comparing aspirin users with aspirin nonusers.

**Table 1 cancers-15-00097-t001:** Demographic and baseline clinical characteristics of the two propensity-matched study cohorts.

	Aspirin User (*n* = 24,030)		Aspirin Nonuser (*n* = 24,030)		Standardized Mean Difference
Characteristics	*n*	%	*n*	%	
**Sex**					
Female	11,256	46.84	11,322	47.12	0.060
Male	12,774	53.16	12,708	52.88	0.060
**Age (years)**					
20–39	1487	6.19	2202	9.16	0.112
40–59	10,314	42.92	9862	41.04	0.038
60–79	10,695	44.51	9587	39.90	0.093
≥80	1534	6.38	2379	9.90	0.129
Mean ± SD	60.34 ± 13.27		60.19 ± 15.04		0.011
**Comorbidities**					
Hypertension	17,307	72.02	17,326	72.10	0.002
Diabetes mellitus	8230	34.25	8195	34.10	0.003
Hyperlipidemia	11,527	47.97	11,698	48.68	0.014
Atrial fibrillation	998	4.15	959	3.99	0.008
Alcohol-related illness	1029	4.28	1065	4.43	0.007
Mild liver disease	8248	34.32	8403	34.97	0.014
Moderate or severe liver disease	186	0.77	174	0.72	0.006
Hepatitis B	1015	4.22	1067	4.44	0.011
Hepatitis C	608	2.53	609	2.53	0.000
Rheumatic disease	1323	5.51	1380	5.74	0.010
Myocardial infarction	924	3.85	899	3.74	0.005
Congestive heart failure	3650	15.19	3623	15.08	0.003
Chronic pulmonary disease	9989	41.57	10,016	41.68	0.002
Peptic ulcer disease	9259	38.53	9377	39.02	0.010
Gastrointestinal bleeding	3063	12.75	3053	12.70	0.001
Renal disease	2270	9.45	2279	9.48	0.001
**Prior drug use**					
Antihypertensive agents	16,767	69.78	16,665	69.35	0.009
Drugs for cardiac therapy	6097	25.37	5789	24.09	0.030
Peripheral vasodilators and vasoprotectives	4846	20.17	4717	19.63	0.013
HMG-CoA reductase inhibitors	3770	15.69	3888	16.18	0.013
Other lipid-modifying agents	1071	4.46	1093	4.55	0.004
Hypoglycemia agents	5111	21.27	5080	21.14	0.003
Coumadin and heparin	1461	6.08	1339	5.57	0.022
Other antithrombotic agents	4702	19.57	4492	18.69	0.022
Proton pump inhibitors	1639	6.82	1686	7.02	0.008
H2-receptor antagonists	5291	22.02	5284	21.99	0.001
Antacids	15,551	64.71	15,715	65.40	0.014
Estrogens and progestogens	864	3.60	850	3.54	0.003
Non-steroidal anti-inflammatory drugs	16,905	70.35	17,606	73.27	0.065
**Duration between ICCD date and index date** **, days (Mean ± SD)**	991 ± 1060		1006 ± 972		0.015

ICCD, ischemic cardiac or cerebrovascular disease; HMG-CoA, 3-hydroxy-3-methyl-glutaryl-coenzyme A; H2-receptor, histamine type-2 receptor; SD, standard deviation. Prior drug use was defined as medications that were prescribed within 6 months prior to the index date. Standardized mean difference ≤ 0.2 indicates a negligible difference between the two study cohorts.

**Table 2 cancers-15-00097-t002:** The dose responses to aspirin among the ischemic cardiovascular and ischemic cerebrovascular disease survivors in our study.

Variables	Cancer (*n* = 2150)	Crude HR			Adjusted HR		
		HR	(95%CI)	*p*-Value	HR	(95%CI)	*p*-Value
**Aspirin not used**	1293	1.00			1.00		
**Aspirin used**							
<1 year	690	0.48	(0.44–0.53)	<0.001	0.43	(0.39–0.47)	<0.001
1–2 years	80	0.47	(0.38–0.59)	<0.001	0.37	(0.30–0.47)	<0.001
2–3 years	46	0.53	(0.39–0.71)	<0.001	0.42	(0.31–0.56)	<0.001
>3 years	41	0.41	(0.30–0.55)	<0.001	0.31	(0.23–0.43)	<0.001

CI, confidence interval; HR, hazard ratio. HR adjusted for aspirin use, age, sex, baseline comorbidities, and drug use in Cox proportional hazard model.

**Table 3 cancers-15-00097-t003:** Incidence rate and risk of developing primary cancer in ICCD survivors stratified by sex and age.

	Aspirin User		Aspirin Nonuser		Aspirin User vs. Aspirin Nonuser	
	Event	IR	Event	IR	Crude HR (95% CI)	Adjusted HR (95% CI)
**Sex**						
Female	298	4.74	446	9.93	0.48 (0.42–0.56) *	0.43 (0.37–0.50) *
Male	559	8.08	847	17.94	0.47 (0.42–0.52) *	0.42 (0.38–0.47) *
**Age, years**						
20–39	15	1.91	23	2.07	0.92 (0.48–1.76)	0.78 (0.39–1.54)
40–59	241	4.18	382	8.77	0.48 (0.41–0.56) *	0.45 (0.38–0.53) *
60–79	532	8.79	772	23.19	0.40 (0.36–0.44) *	0.39 (0.35–0.44) *
≥80	69	11.37	116	27.80	0.47 (0.35–0.63) *	0.43 (0.31–0.58) *

ICCD, ischemic cardiac or cerebrovascular disease; IR, incidence rates (per 1000 person-years); HR, hazard ratio; CI, confidence interval. In the Cox proportional hazard model, adjusted HR was obtained after adjustment for potential confounders, including age, sex, baseline comorbidities, and prior drug use. * *p* < 0.001.

**Table 4 cancers-15-00097-t004:** Incidence rate and risk of developing primary cancer in ICCD survivors stratified by various types of cancer.

Cancer (ICD-9-CM)	Aspirin User		Aspirin Nonuser		Aspirin User vs. Aspirin Nonuser	
	Event	IR	Event	IR	Crude HR (95 % CI)	Adjusted HR (95 % CI)
Head and neck (140–149)	69	0.52	116	1.26	0.44 (0.32–0.59) *	0.42 (0.31–0.57) *
Esophagus (150)	15	0.11	22	0.24	0.51 (0.26–0.98)	0.45 (0.23–0.88)
Stomach (151)	51	0.39	77	0.84	0.49 (0.34–0.70) *	0.43 (0.30–0.62) *
Colon (153, 154)	215	1.63	379	4.11	0.41 (0.35–0.49) *	0.36 (0.31–0.43) *
Hepatoma (155)	154	1.17	190	2.06	0.58 (0.47–0.72) *	0.51 (0.41–0.63) *
Pancreas (157)	31	0.23	28	0.30	0.84 (0.50–1.40)	0.75 (0.45–1.27)
Lung (162)	147	1.11	166	1.80	0.63 (0.51–0.79) *	0.55 (0.44–0.69) *
Breast (174, 175)	59	0.45	98	1.06	0.40 (0.29–0.55) *	0.42 (0.30–0.58) *
Ovary (183)	9	0.07	9	0.10	0.68 (0.27–1.73)	0.75 (0.29–1.93)
Prostate (185)	107	0.81	208	2.26	0.37 (0.30–0.47) *	0.33 (0.26–0.41) *

ICCD, ischemic cardiac or cerebrovascular disease; IR, incidence rates (per 1000 person-years); HR, hazard ratio; CI, confidence interval. In the Cox proportional hazard model, adjusted HR was obtained after adjustment for age, sex, baseline comorbidities, and prior drug use. * *p* < 0.001.

## Data Availability

The data generated in this study are available upon request from the corresponding author.

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
