# Peer review of "Aspirin and Primary Cancer Risk Reduction in Ischemic Cardiac or Cerebrovascular Disease Survivors: A Nationwide Population-Based Propensity-Matched Cohort Study"

_cancers, 2022, doi:10.3390/cancers15010097_

Round 1

Reviewer 1 Report

1. The definition of aspirin user is unclear.  The authors says prescription of aspirin >90 day was defined as aspirin user. What was the time window for the assessment? What if a person is diagnosed with ICCD in 2007, and was prescribed aspirin from 2009? In that case, the index date is some day in 2009. 

The authors also say aspirin nonusers were  assigned a randomly generated date within the same year as index dates. But, in this case, is there asy measure that ICCD was diganosed in 2007?  I feel description of study population is insufficient to ensure the comparability of two groups. In addition, I think diagnosis of ICCD should be the index date, and used in matching to ensure comparabilty. 

The author wrote: To enhance diagnostic validity, patients with at least two consistent  diagnoses of ICCD from outpatient medical records were enrolled (Figure 1). On the basis  of a new diagnosis of ICCD, patients with a prescription of aspirin (> 90 days defined daily  doses) were defined as aspirin users, whereas those without the prescription were defined   as aspirin nonusers. In Taiwan, low-dose aspirin is a prescription drug with 100 mg/day  as the most frequent formulation indicated for CVD prevention [18]. The index dates for  aspirin users were defined as the first date of aspirin prescription. Aspirin nonusers were  assigned a randomly generated date within the same year as index dates. 

2. Relatedly, as time window for aspirin use assessment is not described, exposure-outcome time sequence is unclear. 

3. Aspirin is almostly essential drug for ischemic heart disease during the study period. For example, those who got stent should be on dual antiplatelet therapy (aspirin + clopidogrel) for at least 6 months.  Then, who are the control group? Why some people used aspirin? and others did not? If there is some reasons for this, this cannot be adjusted by propensity score matching. 

Reviewer 2 Report

·       I am concerned about confounding by indication, i.e. that the individuals who are prescribed aspirin have important differences from the individuals who are not prescribed aspirin. Did the individuals who were prescribed aspirin have more severe (or less severe) ICCD? Were the individuals who were prescribed and willing to take aspirin more educated and/or more likely to engage in other healthy behaviors that could also reduce cancer risk? Do you have any information on smoking status, obesity, or physical activity? Without these key confounders, it is hard to say whether aspirin is truly protective, or simply associated with other protective factors.

·       I am also unclear as to how you defined aspirin users and non-users. Were individuals classified as aspirin users if they were prescribed aspirin at any point during follow-up, and non-users if they did not have any prescriptions for aspirin during the entirety of follow-up? If so, there is potential for bias, as individuals with shorter survival time would also have less opportunity to initiate aspirin use.

·       In general, more information on the characteristics of the aspirin users would be helpful. What was the average duration of aspirin use (and how was this determined)? What was the average time between incident ICCD and initiation of aspirin use? Was there any history of aspirin use in the study participants prior to the incident ICCD?

·       The age bins for the propensity score matching were very wide. There could still be confounding by age if there is imbalance within each of the age bins. Could you provide the mean age within each bin, for aspirin users and non-users? And did you adjust for age as a continuous covariate in the multivariable models?

·       The follow-up time for the study is very short (only 5.49 and 3.15 years for aspirin users and non-users, respectively). With a short follow-up period, it is hard to say whether aspirin is associated with reduced risk of cancer, or whether aspirin simply delays cancer detection and diagnosis. This should be discussed as a limitation.

·       What are the characteristics of the aspirin users and non-users who were excluded from the study? (n=14,323 aspirin users and n=29,017 non-users excluded according to Figure 1), and why were these individuals excluded? This could affect the generalizability of the findings.

Minor comments:

·       Introduction – refs #1 and #2 are from U.S.-based organizations. Are similar recommendations utilized in Taiwan?

·       Line 47-49: “…regular long-term aspirin intake has also been used as a chemopreventive approach for primary cancer prevention in the general population.” The word use seems to imply that aspirin has been prescribed for cancer chemoprevention, which is not true. I suggest saying instead that regular long-term aspirin intake has been studied as a potential chemopreventive approach.

·       In the statistical analysis section, please list which covariates were adjusted for in the adjusted Cox proportional hazards models.

·       For the cumulative risk calculations, did you consider using competing risk models? This could be especially important for this study population given that the population is at high risk for additional adverse cardiovascular outcomes.

·       The null associations for ovarian and pancreatic cancer could be due to low power, since these cancer sites had the fewest events.

Reviewer 3 Report

 Manuscript ID

cancers-1938273

Aspirin Decreases Primary Cancer Risk in Ischemic Cardiac or Cerebrovascular Disease Survivors: A Nationwide Population-based Propensity-matched Cohort Study

Brief summary

The authors describe a population-based study exploring the anticancer properties of aspirin in a high-cancer risk population of ICCD survivors. The population is propensity-matched (1:1) for age, sex, comorbidities, prior medications, ICCD diagnosis year, and year of index dates, and the risk of cancer incidence (including by type) is examined. The main finding was that aspirin reduced the risk of cancer incidence in ICCD patients, and in 7 of the top 10 cancer types, with a benefit linked to duration. The current understanding of aspirin and chemoprevention is suitably described and the manuscript well written. The study has suitable ethics and consent.

Reviewers queries:

·        Section 1: The introduction identifies current field of aspirin showing benefit in some studies, and no benefit in others, but the authors do not discuss how these studies differed or why the outcomes were found to differ.  For example: (line 50) “Some studies reported that low-dose aspirin use was associated with a reduced risk of several common types of cancer [3, 7-16], most notably colorectal cancer (CRC) [4, 17, 18]; however, some other studies found no beneficial effects [19-24]”.

And in line 66: “some investigators found that daily use of aspirin resulted in a significant reduction in the long-term risk of CRC or cancer-related death [17, 39, 40], whereas others reported null effects [41, 42].” Please interrogate the prior work in  little more detail.

·        2.1  Data source. The authors state they are using the Taiwan National Health Insurance Research Database (NHIRD) (Line 80), which “provided the entire population in Taiwan a universal health insurance and covered 99% of their health care needs and medical services.”, but then go on to state that they used the longitudinal health insurance database 2000 (LHID 2000). LHID 2000 contains 1,000,000 participants (approximately 5% of Taiwan’s population) randomly selected from National Health Insurance beneficiaries in Taiwan (line 86). It is not clear whether the LHID 2000 dataset is a subset of the NHIRD dataset, and if so (as suspected), what this sub-dataset offers. Figure 1 also describes the NHIRD cohort of 2000-2013 as having 1 million enrolees.  So, is this the whole population during this time frame?

·        NHIRD included ambulatory care, inpatient care, and registration data of the insured (line 83). It is not clear how medication history was captured.

·        Section 2.2: The authors do not actually describe the definition of the Index– “ a new diagnosis of ICCD”?  So, for the patients described in line 95 (patients with at least two consistent diagnoses of ICCD from outpatient medical records) which date is the index date? 

·        Figure 1: shows the patient selection process, although it would have been good to include a further line for the numbers within each of the age categories.

·        2.3 Covariate Assessment – demographic data age and sex.  What about risk factors known to be associated with cancer? E.g. smoking and alcohol use, exercise/activity levels,  past cancer history/family cancer history?

·        Aspirin definition – it is unclear how aspirin use was defined, please describe in greater detail.  Line 96-98 states “On the basis of a new diagnosis of ICCD, patients with a prescription of aspirin (> 90 days defined daily doses) were defined as aspirin users, whereas those without the prescription were defined 98 as aspirin nonusers.”  This definition appears to mean that a single exposure of aspirin, as long as it was for at least 90 days, at some point over the follow up period reported for each group (line 166:  aspirin user follow up = 5.49yrs and non-user groups 3.15 yrs) would suffice as being an aspirin user. Is this correct?  The ‘further analyses’ described in lines 173 – 177 indicates aspirin use for varying durations was examined – where is this data? It would be good to have a better understanding of what the aspirin usage was like in these ICCD survivors, and whether any comparisons were made (aspirin user vs non-user) in a subgroup of individuals who had not experienced ICCD.

·        Did the authors have access to the prescription ‘supply date’ or just ‘prescription’ date?  Using supply date would at least confirm that the patient filled the script.  “Medications that are commonly prescribed to this patient population within 6 months prior to the index date were also identified (labelled as prior drug use).”

·        Medications listed were:  antihypertensive agents, drugs for cardiac therapy, peripheral vasodilators and vasoprotectives, 3-hydroxy-3-methylglutaryl coenzyme A reductase inhibitors, other lipid-modifying agents, hypoglycemia agents, coumadin and heparin, other antithrombotic agents, proton pump inhibitors, histamine type-2 receptor antagonists, antacids, estrogens and progesterone, and nonsteroidal anti-inflammatory drugs. How do you account for NSAIDS purchased without a prescription?  Table 1 shows 70% of the study population used ‘other anti-thrombotic drugs’ – did the authors consider looking at these by class?

·        There is some evidence surrounding metformin as having anticancer properties.  I note 35% of the population was diabetic, with around 21% using hypoglycaemic drugs. Did the authors consider dividing the hypoglycemics agents into specific classes, or isolating at least, metformin and insulin?

·        What was the overall median follow up time? “The follow-up for patients began on the index date and lasted until withdrawal from the National Health Insurance program, until the diagnosis of primary cancer, or on December 31, 2013, whichever was sooner”. 

·        In Results: Table 1 shows categorical variables were presented as n (%) and includes the Standardized mean difference, yet the authors also conducted a Chi -squared test (line 137). Can the authors include the chi-square outcomes in table 1?

·        Patients diagnosed with various common cancers have a higher prevalence of pre-existing CVD than the general population [31], thus promoting an urgent need to reduce the prevalence of primary cancer in this high-risk subpopulation. To this end, taking low-dose aspirin daily in ICCD survivors is not for the sake of primary cancer prevention but for the secondary prevention of ischemic attack.

·        (line 220) “Patients diagnosed with various common cancers have a higher prevalence of pre-existing CVD than the general population [31]” Is this not inconsistent with your message? The point here would be whether or not the pre-existing CVD was being treated.  Thus, please elaborate if in the absence of treatment.

·        (Line 224) “Although several antiplatelet therapies are available for these patients, aspirin remains as the anti-platelet agent of choice in the setting of secondary prevention according to current guidelines [45, 46]. “Have the authors looked at the potential anti-cancer properties of the other therapies used in the secondary prevention setting?

·        Findings for ASPREE should be considered in discussion (lines 242 – 248) given it found harm with aspirin with respect to cancer-related mortality in older adults, and an increased risk of late stage cancer incidence with aspirin.  Whilst ASPREE is referenced in lines 263 -264, the risk for late stage cancer incidence has not been mentioned. This would impact the discussion on lines 265-266.

Round 2

Reviewer 2 Report

Thank you for addressing many of my initial comments. However, some of my major concerns are still not addressed.

My primary concern is that the matching was done incorrectly. Which individuals were considered "non-aspirin users," eligible to be matched to the aspirin-users? If aspirin-users were individuals who used aspirin at any point during the follow-up period, and aspirin non-users were only drawn from individuals who were not prescribed aspirin at any point during follow-up, there is potential for immortal time bias (those who survive longer have more opportunity to initiate aspirin use, which could make aspirin use appear falsely protective).

Incidence density sampling could be used to avoid this bias - i.e., match aspirin users to individuals who were not prescribed aspirin prior to the matched index date (but these "non-aspirin users" could still potentially start using aspirin later during the follow-up period).

Thank you for providing some additional detail on which covariates were adjusted for in the Cox proportional hazards models. However, I am still concerned that there was no adjustment for lifestyle factors such as BMI and smoking history.
